# The Association between Self-Rated Health Status, Psychosocial Stress, Eating Behaviors, and Food Intake According to the Level of Sunlight Exposure in Korean Adults

**DOI:** 10.3390/ijerph20010262

**Published:** 2022-12-24

**Authors:** Hyo-Jeong Hwang, Yean-Jung Choi, Dongwan Hong

**Affiliations:** 1Department of Food and Nutrition, Sahmyook University, Seoul 01795, Republic of Korea; 2Department of Medical Informatics, College of Medicine, Catholic University of Korea, Seoul 06591, Republic of Korea

**Keywords:** sunlight exposure, self-rated health status, psychosocial stress, eating behaviors, food intake

## Abstract

Sunlight exposure has been reported to have various beneficial effects on human health. This study investigated the relationship between self-rated health status, psychosocial stress, eating behaviors, and food intake according to sunlight exposure in 948 adults. Sunlight exposure was classified as less than one hour, less than three hours, and greater than three hours. Of the participants, 49.2% had fewer than three hours of daily exposure to sunlight. Regarding participants exposed to sunlight for less than one hour, the largest response was that they did not engage in outdoor activities on weekdays or weekends, and the rate of being outdoors in the shade on sunny days was the highest in this group at 42.7%. Furthermore, the participants exposed to sunlight for less than one hour had a lower health response than the other two groups, and there were significantly more participants classified in the stress risk group. Regarding eating habits, those with less than an hour of exposure to sunlight frequently ate fried foods, fatty foods, added salt, and snacks, and had significantly lower total dietary scores or three regular meals. Additionally, their frequency of consumption of cereals, milk and dairy products, orange juice, and pork was also significantly lower than the other groups. Thus, it is necessary to provide sufficient guidelines for adequate sunlight exposure and food intake because participants with low sunlight exposure may have low vitamin D synthesis and insufficient food intake.

## 1. Introduction

UV exposure from sunlight is one of the major environmental risk factors for skin cancer [1,2] and various recommendations have been made to block exposure to sunlight. However, recent studies have reported an association with autoimmune diseases, fractures, cancer, cardiovascular disease, and diabetes, which negatively affect health because of active sun protection or insufficient vitamin D intake [3,4,5,6]. There is a growing interest in the health burden caused by insufficient sunlight exposure rather than the risk of excessive sunlight exposure. Stress and other negative emotions such as depression and anxiety can also affect people’s eating behaviors and harm their physical and mental health [7]. Emotional eating can be understood as the behavior of choosing to eat certain foods to deal with positive or negative experiences and is associated with obesity, eating disorders, and other factors. Therefore, investigating this aspect is relevant to reducing these negative outcomes in the population. Recently, some studies have begun to focus on the effects of sunlight on depression; however, the current evidence is still insufficient and existing studies have drawn inconsistent conclusions [8].

A major benefit of sunlight exposure is its role in synthesizing vitamin D in the skin [9,10]. Although there are differences depending on the time of day, season, latitude, and individual sensitivity to sunlight, 3000 international units of vitamin D can be synthesized when an arm or leg is exposed to sunlight for five to ten minutes [11]. Vitamin D levels often do not meet the recommended levels without sufficient sunlight exposure, the main source of vitamin D. Therefore, sunlight exposure should be considered to quantify vitamin D and assess its association with disease. However, since UV radiation is affected by various factors such as altitude, latitude, time of day, season, clouds, and air pollution, measuring sunlight exposure is very complicated [12]. Additionally, sun protective behaviors that affect sun exposure, such as clothing, sunscreens, and sunglasses, should also be considered.

Several previous studies have been conducted on the development of questionnaires for the assessment of sunlight exposure [12,13]. The National Health and Nutrition Examination Survey (NHANES) has been investigating several factors such as: whether skin changes when exposed to sunlight, being in the shade when exposed to sunlight, wearing a long-sleeved shirt, applying sunscreen, how much tanning an individual does, and how much time an individual spends outdoors in the sun on weekdays and weekends [14].

Studies on sunlight exposure, blood vitamin D, and diseases have been conducted in Korea but they are still insufficient [15,16]. According to the blood vitamin D concentration data from the Korean National Health and Nutrition Examination Survey, more than 80% of people are vitamin D deficient [17,18]. However, in most cases only blood levels were measured and there are insufficient basic data to evaluate the effects of sun exposure or vitamin D on health because of a lack of studies related to vitamin D evaluation through food intake or exposure to sunlight.

Therefore, the purpose of this study was to develop and apply related items to measure sunlight exposure and to provide basic data for studies related to sunlight exposure and vitamin D by examining the relationship between the level of sunlight exposure and related behaviors, self-rated health status, psychosocial stress, vitamin-D-related food sources, and eating behaviors in Korean adults.

## 2. Materials and Methods

### 2.1. Study Participants

In this study, a survey was conducted from April 2012 to May 2012, targeting adults over the age of 19 living in Seoul and Gyeonggi Province. The survey was conducted on 1048 participants who completed the research consent form and questionnaire; 948 participants (90.5%) were finally analyzed, excluding 100 participants who delivered insufficient responses.

### 2.2. Primary Outcome Measure: Sunlight Exposure

To measure the sunlight exposure of the participants, the time of exposure to sunlight during the day and outdoor activities from 9:00 am to 5:00 pm on weekdays and weekends during the past 30 days were investigated. Daily sunlight exposure was assessed by asking participants, “What is the average amount of time you are exposed to direct sunlight each day?” The response options were “<1 h”, “1–3 h”, “3–5 h”, “5–7 h”, and “>7 h”. The resulting responses were consequently divided into three categories: “less than 1 h”, “1–3 h”, and “more than 3 h”. Outdoor activity time during the weekdays was answered as a continuous variable (hours) or “did not go outdoors”, whereas outdoor activity time during the weekends was answered as a continuous variable (time), “did not go outdoors”, or “go to school (or go to work) all week”. The resulting responses were consequently divided into three categories: “never”, “less than 3 h”, and “more than 3 h”. Moreover, behavioral factors related to exposure to sunlight, the amount of time spent in the shade when going out on a sunny day, and whether or not sunscreen was applied were measured. Staying in the shade and using sunscreen during outdoor activities were classified as frequent, moderate, and rare. Response options were “never”, “rarely”, “sometimes”, “most of the time”, and “always”.

### 2.3. Secondary Outcome Measure: Self-Rated Health Status, Psychosocial Stress, and Physical Activity

Self-rated health status was measured on a scale ranging from “very good (1 point)” to “very poor (5 points)”. Furthermore, the Korean version of SF-12, an abbreviation of SF-36, was used as a health-related quality of life measurement tool [19]. The SF-12 consists of the following areas: physical function, physical role limitations, pain, overall health, vitality, social functioning, emotional role limitations, and mental health [20].

Psychosocial stress was measured using the Psychosocial Well-being Index-Short Form (PWI-SF) used in the field of health care. It consists of 18 items on social role performance and self-confidence, depression, sleep disorders and anxiety, overall health and life, etc., using a 4-point Likert scale for each item, with higher scores indicating higher levels of stress. According to the PWI-SF score, the high-risk group scored 27 points or more, the potential stress group scored nine to 26 points, and the healthy group scored eight points or fewer [21].

Physical activity was classified as vigorous activity (for more than 10 min), moderate activity (for more than 10 min), and walking activity (walking for more than 10 min). Response options for how many days in a week these were achieved on were “never”, “1 day”, “2 days”, “3 days”, “4 days”, “5 days”, “6 days”, and “everyday”.

### 2.4. Tertiary Outcome Measure: Eating Behaviors and Food Intake

Eating behaviors measured both drinking and smoking status and regularity and balance of meals. This was a simple eating survey that consisted of ten items measuring: dairy intake, protein intake, vegetable intake, fruit intake, fried food intake, fatty meat intake, salty eating or not, meal regularity, snack intake, and balanced food intake. It was measured on a 3-point scale of “always like that,” “normal,” and “not so much,” with a higher score indicating desirable eating behavior practice.

Frequency of food intake was measured for 18 major sources of vitamin D, a key nutrient associated with sunlight exposure: cereal, milk (regular, low-fat/fat-free), lactobacillus beverage, fermented milk, ice cream, cheese, soy milk, orange juice, tuna (including canned food), mackerel, squid/oyster, anchovy, pork (roasted, stir-fried, boiled, etc.), eggs, mushrooms, etc. The frequency of food intake for each food was based on the average intake over the past month and was divided into nine levels (rarely, once a month, 2–3 times a month, 1–2 times a week, 3–4 times a week, 5–6 times a week, once a day, 2 times a day, 3 or more times a day). The amount of intake was divided into three categories: “below the standard,” “standard amount,” and “more than standard”, based on the standard intake. The survey contents were analyzed by calculating the daily intake frequency taking into consideration the standard intake amount.

### 2.5. Other Measurements

The questionnaire in this study was completed through a preliminary survey referring to previous studies, and all questionnaires were self-recorded by the participants. For general characteristics, age, gender, education level, average monthly income, medical history, dietary supplements, and area of residence were investigated. The body mass index (BMI) was calculated by recording height and weight. The smoking status was categorized as current smoker, past smoker, and non-smoke; the passive smoking status into “over 5/week”, “3–4/week”, “1–2/week”, and “never”. The drinking status was categorized into current drinker, past drinker, and non-drinker, based on their alcohol consumption.

### 2.6. Statistical Analysis

All statistical analyzes were performed using SAS version 9.4. Variable characteristics were expressed as mean and standard deviation for continuous variables and frequency and percentage for categorical variables. For analysis of general characteristics, self-rated health status, food intake, etc., according to sunlight exposure, a general linear model (GLM) followed by Duncan’s multiple range test (DMRT) was performed for continuous variables and the χ2-test was performed for categorical variables according to the characteristics of variables. All significance levels were *p* < 0.05.

## 3. Results

### 3.1. General Characteristics of Study Participants According to Exposure to Sunlight

Table 1 shows the demographic characteristics of participants after dividing them according to sunlight exposure. Of the participants, 89.4% were in their 20s and 64.0% were women. The average daily exposure to sunlight was less than one hour for 26.2%, less than three hours for 49.2%, and 24.7% for more than three hours. Approximately half of the participants were exposed to sunlight for 1 to 3 h. Looking at the characteristics according to sunlight exposure, the proportion of men, specifically those in their 20s, scored significantly higher in the exposure to sunlight for more than one hour category (*p* < 0.001). As for the income level, 37.3% of those who received less than KRW 3,000,000 per month were exposed to sunlight for less than one hour; of those with the highest income level, 36.7% received more than three hours of exposure to sunlight, a significant increase (*p* < 0.05). The largest groups of participants who responded that they had a disease were those exposed to sunlight for less than three hours at 26.5%, followed by 22.8% for three hours or more of sunlight exposure, and 18.2% for less than one hour of sunlight. The proportion of participants who used alcohol was 69.1% and 67.9% in the more than three hours and less than three hours groups, respectively, which was significantly higher than the 55.7% in the less than one hour group (*p* < 0.001). However, there were no statistically significant differences among participants according to the exposure time to sunlight, in obesity, educational level, intake of nutritional supplements, and smoking status.

### 3.2. Behaviors Related to Sunlight Exposure

The results of sunlight exposure and sun protection behaviors are presented in Table 2. Among the participants who were exposed to sunlight for less than one hour during outdoor activities from 9:00 am to 5:00 pm from Monday to Friday, 40.1% of the participants responded that they did not go outdoors during sunny hours. Among the participants exposed to sunlight for more than three hours, 35.2% of the participants reported that they had three hours of outdoor exposure during the week, and the number of participants who spent more time outdoors than the other two groups was significantly higher (*p* < 0.001). Moreover, similar to the results for weekdays, among the participants who responded that they were exposed to sunlight for more than three hours on weekends, the participants who responded that they were outdoors for more than three hours on weekends were the largest group. Among the participants exposed to sunlight for less than one hour, the rate of not engaging in outdoor activities was 42.9%, which was significantly higher than that of the other two groups (*p* < 0.001).

Among the participants with less than one hour of exposure to sunlight, 42.7% of the participants stated that they were in the shade when going out in sunny weather as sun protection behavior. This indicated a relatively high trend compared with 31.3% of the participants exposed to sunlight for more than three hours (*p* < 0.05). However, more than half of the participants were using sunscreen, and there was no difference in the use of sunscreen according to the time of exposure to sunlight. In the past year, 17.3% of tanning experiences were not related to sun exposure.

### 3.3. Self-Rated Health Status, Psychosocial Stress, and Physical Activity According to Sunlight Exposure

The self-rated health status according to sunlight exposure is shown in Table 3. A total of 46.6% of the participants exposed to sunlight for less than one hour responded that they had good health status, whereas more than 60% of participants exposed to sunlight for more than one hour responded that they had good health status. However, there was no significant difference based on sunlight exposure in the self-rated health status score measured through SF-12. The psychosocial stress score was classified into healthy group (<9), potentially stressed group (9–26.9), and high-risk group (≥27). Among the participants exposed to sunlight for less than one hour, the high-stress group was 22.7%, which was significantly higher than that of the other two groups (*p* < 0.05). When comparing the psychosocial stress score according to sunlight exposure as a continuous variable, it was found that the participants with the least exposure to sunlight had a significantly higher stress score than the participants with relatively more exposure to sunlight (*p* < 0.05). Regarding physical activity, participants exposed to sunlight for more than three hours had significantly more active days of both vigorous and moderate physical activity than those in the less than one hour group (*p* < 0.001). Moreover, the number of days of walking for ten minutes or more once a week was also higher in the participants exposed to sunlight for three hours or more group compared with less than one hour group.

### 3.4. Eating Behaviors and Food Intake According to Sunlight Exposure

Table 4 shows the results of eating behaviors and sunlight exposure evaluated by the 10-item dietary questionnaire. Participants with less than one hour of exposure to sunlight had significantly higher fried and oily food intake scores than the other two groups that had relatively longer exposure to sunlight (*p* < 0.05). It was also found that salt was added to food and that snacking was frequent in this group (*p* < 0.05). In contrast, participants who responded that they were exposed to sunlight for more than three hours showed significantly higher scores than the other two groups for the question of eating regularly, and the total score of eating behaviors was also significantly higher, indicating good eating habits. Regarding sunlight exposure by classifying vitamin-D-related food intake into eight food groups (Table 5), the frequency of intake of cereal, milk and dairy products, orange juice, pork, etc., was significantly higher among participants exposed to sunlight for more than three hours. Subsequently, the frequency of intake was found to be low in participants with less than one hour exposure to sunlight.

## 4. Discussion

This study evaluated self-rated health status and sociopsychological stress, mental health statuses associated with exposure to sunlight, in 948 adults, and identified the relationship between eating behaviors and food intake.

The average daily sunlight exposure time was between one and three hours for approximately half of the participants, and 26% of the participants were exposure to sunlight for less than one hour a day. A study on Australians found that adults aged 25–34 had lower levels of physical activity and reported that these lower levels of physical activity may be associated with reduced time spent outdoors. Vitamin D deficiency has also been examined and seasonal differences in vitamin D deficiency were observed [22]. Participants who stated the longest exposure to sunlight were predominantly men, specifically those in their 20s, those with increased income levels, and those who reported that they had a disease. It is thought that the more individuals experienced a disease, the higher their interest in health and the greater their interest in health-related physical activity. However, there were no statistically significant differences among the participants regarding obesity, education level, nutritional supplement intake, and smoking status according to exposure to sunlight.

Regarding outdoor activities during hours of exposure to sunlight on weekdays, 40.1% of participants that had less than an hour of exposure to sunlight stated that they did not engage in outdoor activities and the rate of not engaging in outdoor activities was 42.9% on weekends, which was significantly higher than that in the other two groups. The proportion of participants who reported that they spent most time in the shade when outdoors as a measure to block the sunlight, was also higher in participants with short periods of exposure to sunlight, however, more than half of the participants used sunscreen and did not tan. The proportion of participants tanning in the past year was 17.3%, and this was not related to sunlight exposure. A Canadian study found that 66% of participants stated that they occasionally used sunscreen, whereas the US National Health and Nutrition Survey found that less than 40% of 18–29-year-olds used sunscreen in 2010, and 35% were in the shade when outdoors. It was found that the use of sunscreen was decreased compared with this study, whereas the shade-seeking behavior was similar to this study [23]. A recent Australian study showed widespread and increased use of sunscreen among adults, and physical activity was associated with lower shade-seeking behavior but was associated with higher odds of other sun protection behaviors, such as sun protective clothing and accessories [24,25]. In another study of Australian adults, those who met the physical activity guidelines were more likely to apply sunscreen while outdoors, whereas those who did not meet the guidelines were more likely to seek shade [26,27]. Shade-seeking behavior is influenced by a variety of factors other than sunscreen use, and further research may investigate whether longer periods of physical activity explain higher health perceptions and sun-protective behaviors.

Few participants reported that the shorter their exposure to sunlight, the better their health. As a result of categorizing psychosocial stress scores into the healthy group (<9), potential stress group (9–26.9), and high-risk group (≥27), the stress risk of participants exposed to sunlight for less than one hour was 22.7%. In this study, approximately 59% of participants reported that they were subjectively healthy, however, in the 6^th^ Korea National Health and Nutrition Examination Survey, the reported health was lower than in this study, where 45.6% of the under-30 age group reported that they were subjectively healthy [28]. Regarding the psychosocial stress index, the study period for this study was from April 2012 to May 2012. The stress scores were higher considering the midterm exam period for college students. Several studies have shown that students’ body weight and perceived stress increase whereas their physical activity decreases during exam time. In particular, during exams students showed an increased tendency to eat high energy intake and unhealthy diets [29]. Self-rated health status or the psychosocial stress index, were positively related to sunlight exposure; studies have investigated sunlight’s protective effects against depression [30]. The longer the exposure time to sunlight, the more intensely or moderately physically active individuals are during physical activity. Therefore, measurements of sunlight exposure and physical activity should be considered together. Although there is growing evidence linking environmental factors and depression, highlighting the effect of daylight duration on depression, the literature is still limited and inconclusive [31]. In a study of the effects of sunlight on patients with major depressive disorder, long-term increases in sunlight reduced depression, but the effect was only significant for a limited period [32]. A recent cross-sectional study in China, found that reduced sunlight exposure time was associated with an increased prevalence of depressive symptoms in women over 60 years of age and showed that men were more vulnerable to shorter daylight hours [33]. A Spanish study also found an increased risk of depression in men who lived in areas with shorter daylight hours and found that those under 45 years of age were more sensitive to shorter daylight hours. It has been reported that this may be because young people spend more time working indoors and less time participating in outdoor activities [34].

Regarding eating habits and food intake, it was found that the participants who were exposed to sunlight for less than one hour ate fried and oily food, added salt to food, and consumed snacks more frequently than the other two groups. In contrast, the group exposed to sunlight for a longer duration ate three meals more regularly and had a higher total dietary score, indicating good eating habits. Recently, various studies have reported significant associations between emotional distress management, eating disorder behavior, and weight status [29]. Food consumption is considered an important mood-regulating behavior, and in this context, ’emotional eating’ was used to reflect a tendency to eat in response to emotions rather than hunger or satiety. Indeed, it seems that some individuals are more susceptible to unhealthy changes in food choices, consuming sweeter, saltier, higher-fat, and energy-dense foods to cope with negative emotions. In a recent Brazilian study [7,35], the low self-efficacy for emotional eating found among women of high socioeconomic status was attributable to their greater purchasing power; they had greater access to a wider variety of foods, including highly processed, fat- and sugar-rich products. Eating and nutritional processes are influenced not only by biological factors but also by psychological and cultural factors such as emotions, memory, habits, social status, and many other factors. Studies speculate that low self-efficacy or lack of control in stressful situations may contribute to increased consumption of such products, but no studies have yet directly compared them. Therefore, future research should be conducted to identify the relationship between economic level and emotional eating habits.

The association between depression and vitamin D status due to lack of sun exposure is well established, and dietary and lifestyle approaches have potential as new interventions for managing depressive symptoms. Previous studies have shown an association between an adequate diet and sensible sun exposure for vitamin D deficiency in depressed patients [36]. For vitamin-D-related food intake, cereal, milk and dairy products, orange juice, and pork intake were significantly higher among participants exposed to sunlight for more than three hours. In a study conducted in the UK, vitamin D intake from food, supplements, fish, and shellfish intake were significantly lower in the group with low blood vitamin D levels and summer outdoor activities, normal outdoor activities, and other vigorous physical activity were also reported to be short [37]. This may be explained by the fact that low doses of vitamin D in solar ultraviolet light are associated with low levels of physical activity and poor lifestyles in adults with depression and depressive disorders.

Studies have shown that sunlight also has beneficial psychological effects [30]. A person’s vitamin D levels are greatly affected by sun exposure. A recent cross-sectional study in China found that older adults with low vitamin D serum levels were more likely to show depressive symptoms [33]. Similarly, higher levels of sun exposure were associated with fewer depressive symptoms in middle-aged Australians [38]. In another cross-sectional study, sun exposure was associated with lower scores on the Center for Epidemiologic Research Depression Scale (CES-D) in Chinese college students [39]. The review article also reported that longer exposure to sunlight lowered the risk of depressive symptoms [40]. The clinical significance and causal nature of the reported associations require further research.

In this study, there was no statistically significant difference in BMI associated with the amount of sunlight exposure among the participants; however, it is known that a lack of sunlight can reduce vitamin D production and that this is related to the etiology and seasonality of depression [8,41]. Cross-sectional and prospective studies suggest that emotional eating is one pathway linking depression with weight gain and obesity [41,42,43]. Epidemiological evidence consistently supports the idea that self-reported emotional eating leads to more frequent consumption of energy-dense foods and an increased risk of developing obesity [44]. In this study, participants who also reported the most exposure to sunlight were predominantly male, particularly those in their 20s. Previous studies have reported that stress may contribute differently to weight gain in men compared to women because of the different effects of stress on endogenous hormonal regulators of food intake and energy utilization [45]. Another recent study of stress and eating behaviors among college freshmen reported gender differences in stress-eating pattern associations, with greater stress being a stronger predictor of unhealthy food intake in men compared to women [46]. Additionally, several studies have reported gender differences in hormonal stress response, manifested by increases in circulating free cortisol, a biomarker of hypothalamic-pituitary–adrenal axis activation [47].

This study investigated the relationship between exposure to sunlight and mental health, physical activity, and diet. It was found that the shorter the exposure time to sunlight, the less participants felt that their health status was good and their mental health status was relatively weak because of high psychosocial stress. Additionally, regarding physical activity, shorter exposure to sunlight was associated with less physical activity and increased sun protection behavior; eating habits were also found to be poor. Therefore, in studies related to mental health or vitamin D levels, exposure to sunlight must be investigated and the development of validated evaluation items is required. Furthermore, as individuals with low exposure to sunlight appear to have poor overall health and eating habits, additional research is required and guidelines for correct sunlight exposure and food intake are needed.

## 5. Conclusions

The purpose of this study was to investigate the relationship between self-rated health status, psychosocial stress, eating behaviors, and food intake according to sunlight exposure in 948 Korean adults. In this study, it was found that the less time they were exposed to sunlight, the less likely the participants were to feel that their self-rated health status was good and the higher their psychosocial stress. Furthermore, participants with shorter sunlight exposure exhibited more sun protection behaviors during outdoor activities and were less physically active. They also had lower overall dietary total scores and lower vitamin D intake. Therefore, participants who are less exposed to sunlight have low vitamin D synthesis through the skin and may have insufficient dietary intake, which may be involved with negative health effects. Excessive sunlight exposure is also a problem, however, participants with low sunlight exposure will need adequate sunlight exposure and proper guidelines for food intake. Through this study, it is possible to evaluate the concept of health status and psychosocial stress according to sunlight exposure, so it can expand the perspective of experts in the field and contribute to the development of more efficient health protocols and dietary interventions. Because this study is cross-sectional, it is not possible to demonstrate a subsequent causal relationship between independent variables (e.g., health status, stress, eating behavior) and dependent variables (sunlight exposure). Therefore, a follow-up longitudinal study is needed to establish a causal relationship.

## Figures and Tables

**Table 1 ijerph-20-00262-t001:** General characteristics by sunlight exposure.

	All		Time of Sun Exposure (per Day)						
			<1 h (n = 247)		1–3 h (n = 464)		≥3 h (n = 233)		*p*
	n	(%)	N	(%)	n	(%)	n	(%)	
All	944	(100.0)	247	(26.2)	464	(49.2)	233	(24.7)	
Age									
<30 years	843	(89.4)	195	(79.0)	432	(93.3)	216	(92.7)	<0.0001
≥30 years	100	(10.6)	52	(21.0)	31	(6.7)	17	(7.3)	
Age (years)			26.7	±8.3 ^a^	23.8	±5.6 ^b^	24.4	±8.3 ^b^	<0.0001
Gender									
Male	340	(36.0)	62	(25.1)	184	(39.7)	94	(40.3)	0.000
Female	604	(64.0)	185	(74.9)	280	(60.3)	139	(59.7)	
BMI (kg/m^2^)									
BMI (kg/m^2^) [mean ± SD]			21.0	±2.9	20.8	±2.7	21.1	±2.7	0.518
Education level									
≤12 years	274	(29.0)	71	(28.4)	144	(31.0)	59	(25.3)	0.291
>12 years	670	(71.0)	176	(71.3)	320	(69.0)	174	(74.7)	
Income (10^6^ won/mo)									
<300	296	(32.6)	90	(37.3)	132	(30.0)	74	(32.7)	0.036
300–399	205	(22.6)	57	(23.7)	102	(23.2)	46	(20.4)	
400–499	130	(14.3)	31	(12.9)	76	(17.3)	23	(10.2)	
≥500	276	(30.4)	63	(26.1)	130	(29.6)	83	(36.7)	
Disease									
Yes	221	(23.4)	45	(18.2)	123	(26.5)	53	(22.8)	0.044
No	723	(76.6)	202	(81.8)	341	(73.5)	180	(77.3)	
Dietary supplement									
No	549	(58.2)	145	(58.7)	260	(56.0)	144	(61.8)	0.339
Yes	395	(41.8)	102	(41.3)	204	(44.0)	89	(38.2)	
Smoking									
Non-smoker	738	(83.5)	206	(84.1)	389	(84.6)	188	(80.7)	0.556
Past smoker	63	(6.7)	16	(6.5)	26	(5.7)	21	(9.0)	
Current smoker	92	(9.8)	23	(9.4)	45	(9.8)	24	(10.3)	
Passive smoking									
Never	285	(30.9)	85	(34.8)	140	(30.9)	60	(26.8)	0.163
1–2/week	321	(34.9)	91	(37.3)	157	(34.7)	73	(32.6)	
3–4/week	172	(18.7)	36	(14.8)	88	(19.4)	48	(21.4)	
Over 5/week	143	(15.5)	32	(13.1)	68	(15.0)	43	(19.2)	
Drinking									
Non-drinker	215	(22.8)	82	(33.3)	90	(19.4)	43	(18.5)	0.000
Past drinker	115	(12.2)	27	(11.0)	59	(12.7)	29	(12.5)	
Current drinker	613	(65.0)	137	(55.7)	315	(67.9)	161	(69.1)	

* *p* value was derived from ANOVA or chi-square test. Values not sharing a common or same letter (a, b) and they differ significantly at *p* < 0.05 (Duncan’s multiple range test).

**Table 2 ijerph-20-00262-t002:** Sunlight exposure behaviors.

	All		Time of Sun Exposure (per Day)						
			<1 h (n = 247)		1–3 h (n = 464)		≥3 h (n = 233)		*p*
	n	(%)	N	(%)	n	(%)	n	(%)	
Time outdoors (Monday–Friday, 9 am–5 pm)									
Never	251	(26.6)	99	(40.1)	111	(24)	41	(17.6)	<0.0001
<3 h	571	(60.6)	140	(56.7)	321	(69.3)	110	(47.2)	
≥3 h	121	(12.8)	8	(3.2)	31	(6.7)	82	(35.2)	
Time outdoors (Saturday–Sunday, 9 am–5 pm)									
Never	323	(34.2)	106	(42.9)	153	(33)	64	(27.5)	<0.0001
<3 h	391	(41.4)	106	(42.9)	204	(44)	81	(34.8)	
≥3 h	144	(15.3)	12	(4.9)	66	(14.2)	66	(28.3)	
School or working	86	(9.1)	23	(9.3)	41	(8.8)	22	(9.4)	
Stay in shade									
Frequent	347	(36.8)	105	(42.7)	169	(36.5)	73	(31.3)	0.050
Moderate	372	(39.5)	81	(32.9)	184	(39.7)	107	(45.9)	
Rare	223	(23.7)	60	(24.4)	110	(23.8)	53	(22.8)	
Use sunscreen									
Frequent	487	(51.6)	124	(50.2)	244	(52.6)	119	(51.1)	0.682
Moderate	143	(15.2)	44	(17.8)	63	(13.6)	36	(15.5)	
Rare	314	(33.3)	79	(32)	157	(33.8)	78	(33.5)	
Tanning or burning									
Yes	163	(17.3)	43	(17.6)	89	(19.2)	31	(13.3)	0.153
No	779	(82.7)	202	(82.5)	375	(80.8)	202	(86.7)	

* *p* value was derived from chi-square test.

**Table 3 ijerph-20-00262-t003:** Self-rated health status, psychosocial stress, and physical activity according to sunlight exposure.

	Time of Sun Exposure (per Day)						*p*
	<1 h (n = 247)		1–3 h (n = 464)		≥3 h (n = 233)		
	N	(%)	n	(%)	n	(%)	
Self-rated health status							
Good	115	(46.6)	293	(63.3)	144	(61.8)	0.000
Fair	108	(13.7)	145	(31.3)	73	(31.3)	
Poor	24	(9.7)	25	(5.4)	16	(6.9)	
Stress							
Healthy (<9)	21	(8.5)	56	(12.1)	33	(14.2)	0.034
Potentially stressed (9–27)	170	(68.8)	340	(73.3)	164	(70.4)	
High-risk (≥27)	56	(22.7)	68	(14.7)	36	(15.5)	
Total score of stress (max = 54)	20.0	±8.5 ^a^	18.0	±8.2 ^b^	17.5	±8.5 ^b^	0.002
Total score of SF-12 (max = 100)	67.9	±0.9	67.6	±7.0	67.7	±7.5	0.924
Self-rated health status	2.53	±0.81 ^a^	2.29	±0.76 ^b^	2.30	±0.83 ^b^	0.004
Physical activity							
Vigorous activity (days)	1.30	±1.63 ^b^	1.94	±1.86 ^a^	2.20	±2.12 ^a^	0.000
Moderate activity (days)	1.33	±1.72 ^b^	1.83	±1.93 ^a^	2.02	±2.15 ^a^	0.000
Walking activity (days)	4.89	±2.15 ^b^	5.71	±0.81 ^a^	5.82	±1.87 ^a^	<0.0001

* *p* value was derived from ANOVA or chi-square test. Values not sharing a common or same letter (a, b) and they differ significantly at *p* < 0.05 (Duncan’s multiple range test).

**Table 4 ijerph-20-00262-t004:** Dietary behaviors according to sunlight exposure.

	Time of Sun Exposure (per Day)						
	<1 h (n = 247)		1–3 h (n = 464)		≥3 h (n = 233)		*p*
	Mean	±SD	Mean	±SD	Mean	±SD	
Milk & dairy products	2.17	±0.77	2.06	±0.82	2.12	±0.76	0.194
Meat, fish, egg, bean, or tofu	1.98	±0.72	1.89	±0.74	1.9	±0.72	0.267
Vegetable & kimchi in every meal	1.91	±0.75	1.84	±0.74	1.83	±0.74	0.375
Fruit or fruit juice everyday	2.12	±0.79	2.05	±0.78	2	±0.78	0.254
Fried or stir-fried food	1.95	±0.71 ^a^	1.8	±0.71 ^b^	1.78	±0.7 ^b^	0.011
Fatty meat	2.19	±0.74 ^a^	1.97	±0.75 ^b^	1.9	±0.72 ^b^	<0.0001
Add table salt or soy sauce to food before eating	2.5	±0.64 ^a^	2.46	±0.69 ^a^	2.27	±0.76 ^b^	0.000
Eat three regular meals a day	2.06	±0.76 ^b^	2.08	±0.79 ^b^	2.22	±0.76 ^a^	0.046
Ice cream, cake, snack or soft drink	1.93	±0.76 ^a^	1.83	±0.76 ^a,b^	1.76	±0.76 ^b^	0.049
Variety of foods	1.78	±0.72	1.69	±0.72	1.74	±0.72	0.269
Total score	19.4	±3.0 ^b^	19.5	±3.0 ^b^	20.1	±3.0 ^a^	0.032

* *p* value was derived from ANOVA. Values not sharing a common or same letter (a, b) and they differ significantly at *p* < 0.05 (Duncan’s multiple range test).

**Table 5 ijerph-20-00262-t005:** Food intake according to sunlight exposure.

	Time of Sun Exposure (per day)						
	<1 h (n = 247)		1–3 h (n = 464)		≥3 h (n = 233)		*p*
	Mean	±SD	Mean	±SD	Mean	±SD	
Cereals	0.09	±0.18 ^b^	0.14	±0.28 ^a^	0.17	±0.43 ^a^	0.014
Milk & dairy foods	1.17	±1.08 ^b^	1.34	±0.28 ^a,b^	1.50	±1.37 ^a^	0.018
Soy milk	0.10	±0.22	0.14	±0.27	0.12	±0.20	0.156
Orange juice	0.12	±0.22^c^	0.18	±0.31 ^b^	0.23	±0.41 ^a^	0.001
Fishes	0.45	±0.54	0.51	±0.67	0.53	±0.57	0.281
Pork	0.39	±0.47 ^b^	0.44	±0.52 ^a,b^	0.51	±0.49 ^a^	0.043
Eggs	0.35	±0.43	0.35	±0.39	0.33	±0.32	0.831
Mushroom	0.23	±0.37	0.27	±0.38	0.29	±0.37	0.158

* *p* value was derived from ANOVA. Values not sharing a common or same letter (a, b) and they differ significantly at *p* < 0.05 (Duncan’s multiple range test).

## Data Availability

The data used to support this study is included in the article.

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
