# Peer review of "The Association between Self-Rated Health Status, Psychosocial Stress, Eating Behaviors, and Food Intake According to the Level of Sunlight Exposure in Korean Adults"

_ijerph, 2022, doi:10.3390/ijerph20010262_

Round 1

Reviewer 1 Report

I congratulate the authors for tackling an ambitious and complex piece of research.

This complexity lies in the rigorous management of the study variables: exposure to sunlight, stress, diet.... In the methodology section, it is not clear how these variables are dealt with or how they are analysed.

In the discussion section there are very few references to similar studies. In the conclusions section, the information is repeated in the last paragraph.

There are many bibliographical references more than 10 years old.

I encourage the authors to review these aspects that I point out.

Kind regards

Author Response

Reviewer #1:

  1. I congratulate the authors for tackling an ambitious and complex piece of research. This complexity lies in the rigorous management of the study variables: exposure to sunlight, stress, diet… In the methodology section, it is not clear how these variables are dealt with or how they are analysed.

Response: Thank you very much for your suggestion. Following the reviewer’s suggestion, we have added how the study variables are dealt with or how they are analyzed in the Materials and Methods section.

Page 2 Line 85: Daily sunlight exposure was assessed by asking participants, "What is the average amount of time you are exposed to direct sunlight each day?" The response options were "<1 hour", "1–3 hours", "3–5 hours", "5–7 hours", and ">7 hours". The resulting re-sponses were consequently divided into three categories: "less than 1 hour", "1–3 hours", and "more than 3 hours". Outdoor activity time during the weekdays was answered as a continuous variable (hours) or “did not go outdoors”, whereas outdoor activity time during the weekends was answered as a continuous variable (time), “did not go out-doors”, or “go to school (or go to work) all week”. The resulting responses were conse-quently divided into three categories: "never", "less than 3 hours", and "more than 3 hours".

Page 3 Line 96: Staying in the shade and using sunscreen during outdoor activities were classified as frequent, moderate, and rare. Response options were "never", "rarely", "sometimes", "most of the time", and "always".

Page 3 Line 99: …, psychosocial stress, and physical activity

Page 3 Line 113: Physical activity was classified as vigorous activity (for more than 10 minutes), moderate activity (for more than 10 minutes), and walking activity (walking for more than 10 minutes). Response options for how many days in a week these were achieved on were "never", "1 day", "2 days", "3 days", "4 days", "5 days", "6 days", and "everyday".

Page 3 Line 139: …, dietary supplements, …

Page 3 Line 140: The smoking status was categorized as current smoker, past smoker, and non-smoke; the passive smoking status into "over 5/week", "3–4/week", "1–2/week", and "never". The drinking status was categorized into current drinker, past drinker, and non-drinker, based on their alcohol consumption.

  1. In the discussion section there are very few references to similar studies. In the conclusions section, the information is repeated in the last paragraph.

Response: Thank you for pointing this out. We have carefully revised the Discussion section and have cited relevant references in the discussion to support the inferences of the implications of the results. We also have now corrected the last paragraph repeated in the Conclusion section.

  1. There are many bibliographical references more than 10 years old.

Response: Following the reviewer's suggestion, the references older than 10 years have been replaced or added with recently reported articles.

Reviewer 2 Report

Thank you for giving me an opportunity to review this empirical study pertaining to the relationships between sunlight exposure and behavioral/health outcomes among Korean adults. This paper pointed out the significancy of sufficient sunlight exposure which could be associated with better health behaviors and self-rated health status. The authors lay out the arguments clearly, while the contributions are mainly on the descriptive level. There are several problems that the authors should carefully address:

1. As a quantitative study, this article does not have a clear theoretical framework. It is necessary to clarify the casual relationships among these “primary/secondary/tertiary outcome” variables.

2. The part of literature review is inadequate and fails to find out the shortcomings of existing research. Although the authors cited some references, this paper lacked a thorough review of studies related to the possible outcomes of sunlight exposure, e.g., the non-linear effects of sunlight exposure upon individual health.

3. In the section of methodology, the authors mentioned that “a general linear model (GLM) was performed for continuous variables”. However, the following tables only presented bivariate analysis of variance. It is suggested to supplement multivariate regression analysis to identify the effect size of sunshine duration on health outcomes.

In summary, this paper is well organized and provides some novel findings. Nevertheless, there are questions regarding the clarification of theoretical framework and analytical strategy.

Author Response

Reviewer #2:

Thank you for giving me an opportunity to review this empirical study pertaining to the relationships between sunlight exposure and behavioral/health outcomes among Korean adults. This paper pointed out the significancy of sufficient sunlight exposure which could be associated with better health behaviors and self-rated health status. The authors lay out the arguments clearly, while the contributions are mainly on the descriptive level. There are several problems that the authors should carefully address:

  1. As a quantitative study, this article does not have a clear theoretical framework. It is necessary to clarify the casual relationships among these “primary/secondary/tertiary outcome” variables.

Response: Thank you for this advice and we do agree that this information is needed for improving the contextual content.

Page 1 Line 12: Sunlight exposure has been reported to have various beneficial effects on human health.

Page 1 Line 38: Stress and other negative emotions such as depression and anxiety can also affect people's eating behaviors and harm their physical and mental health [7]. Emotional eating can be understood as the behavior of choosing to eat certain foods to deal with positive or negative experiences and is associated with obesity, eating disorders, and other fac-tors. Therefore, investigating this aspect is relevant to reducing these negative out-comes in the population. Recently, some studies have begun to focus on the effects of sunlight on depression; however, the current evidence is still insufficient and existing studies have drawn inconsistent conclusions [8].

  1. The part of literature review is inadequate and fails to find out the shortcomings of existing research. Although the authors cited some references, this paper lacked a thorough review of studies related to the possible outcomes of sunlight exposure, e.g., the non-linear effects of sunlight exposure upon individual health.

Response: Following the reviewer’s suggestion, we tried to revise the Discussion section with careful consideration. We have now included the adequate references to review of studies related to the possible outcomes of sunlight exposure.

  1. In the section of methodology, the authors mentioned that “a general linear model (GLM) was performed for continuous variables”. However, the following tables only presented bivariate analysis of variance. It is suggested to supplement multivariate regression analysis to identify the effect size of sunshine duration on health outcomes.

Response: Thank you for your suggestion. We have added a table for this information in the Supplementary material (Table S1). In the table, we present the F values and p values of health variables including self-rated health status, psychosocial stress, and eating behaviors according to the time of sunlight exposure per day among participants.

In summary, this paper is well organized and provides some novel findings. Nevertheless, there are questions regarding the clarification of theoretical framework and analytical strategy.

Round 2

Reviewer 1 Report

Thank you for incorporating the proposals I sent you. The article has been improved in the areas noted.

Good work!

Best regards